# CAiRE-COVID: A Question Answering and Query-focused Multi-Document Summarization System for COVID-19 Scholarly Information Management

**Dan Su**[1,2], **Yan Xu**[1], **Tiezheng Yu**[1], **Farhad Bin Siddique**[1,2],
**Elham J. Barezi**[1] , **Pascale Fung**[1,2]

[1]Center for Artificial Intelligence Research (CAiRE)
The Hong Kong University of Science and Technology, Clear Water Bay, Hong Kong
[2] EMOS Technologies Inc.
{dsu,yxucb,tyuah,fsiddique,ejs}@connect.ust.hk
pascale@ece.ust.hk

## Abstract

We present CAiRE-COVID, a real-time question answering (QA) and multi-document summarization system, which won one of the 10 tasks in the Kaggle COVID-19 Open Research Dataset Challenge[1], judged by medical experts. Our system aims to tackle the recent challenge of mining the numerous scientific articles being published on COVID-19 by *answering* high priority questions from the community and *summarizing* salient question-related information. It combines information extraction with state-of-the-art QA and query-focused multi-document summarization techniques, selecting and highlighting evidence snippets from existing literature given a query. We also propose query-focused abstractive and extractive multi-document summarization methods, to provide more relevant information related to the question. We further conduct quantitative experiments that show consistent improvements on various metrics for each module. We have launched our website CAiRE-COVID[2] for broader use by the medical community, and have open-sourced the code[3] for our system, to bootstrap further study by other researches.

## 1 Introduction

Since the COVID-19 outbreak, a huge number of scientific articles have been published and made publicly available to the medical community (such as bioRxiv, medRxiv, WHO, pubMed). At the same time, there are emerging requests from both the medical research community and wider society for efficient management of the information about COVID-19 from this huge number of research articles. High priority scientific questions need to be *answered*, e.g., *What is known about transmission, incubation, and environmental stability? What do we know about COVID-19 risk factors?* and *What do we know about virus genetics, origin, and evolution?* Furthermore, question-related salient information needs to be *summarized*, so that the community can digest important contextual information more efficiently and keep up with the rapid acceleration of the coronavirus literature.

The release of the COVID-19 Open Research Dataset (CORD-19)[1] (Wang et al., 2020), which consists of over 158,000 scholarly articles about COVID-19 and related coronaviruses, creates an opportunity for the natural language processing (NLP) community to address these requests. However, it also poses a new challenge since it is not easy to extract precise information regarding given scientific questions and topics from such a large set of unlabeled resources.

To meet the requests and challenges for scholarly information management related to COVID-19, we propose CAiRE-COVID, a neural question answering and query-focused multi-document summarization system. Given a user query, the system first *selects* the most relevant documents from the CORD-19 dataset[1] with high coverage via a Document Retriever module. It then *highlights* the answers or evidence (text spans) for the query, given the relevant paragraphs, by a Snippet Selector module via question answering (QA) models. Furthermore, to efficiently *present* COVID-19 question-related information to the user, we propose a query-focused Multi-Document Summarizer to generate abstractive and extractive summaries related to the question, from multiple retrieved answer-related paragraph snippets. We leverage the power of the generalization ability of pre-trained language models (Lewis et al., 2019; Yang et al., 2019; Lee et al., 2020; Su et al., 2019) by fine-tuning them for QA and summarization, and propose our own adaptation methods for the COVID-19 task.

---

[1]https://www.kaggle.com/allen-institute-for-ai/CORD-19-research-challenge
[2]https://caire.ust.hk/covid

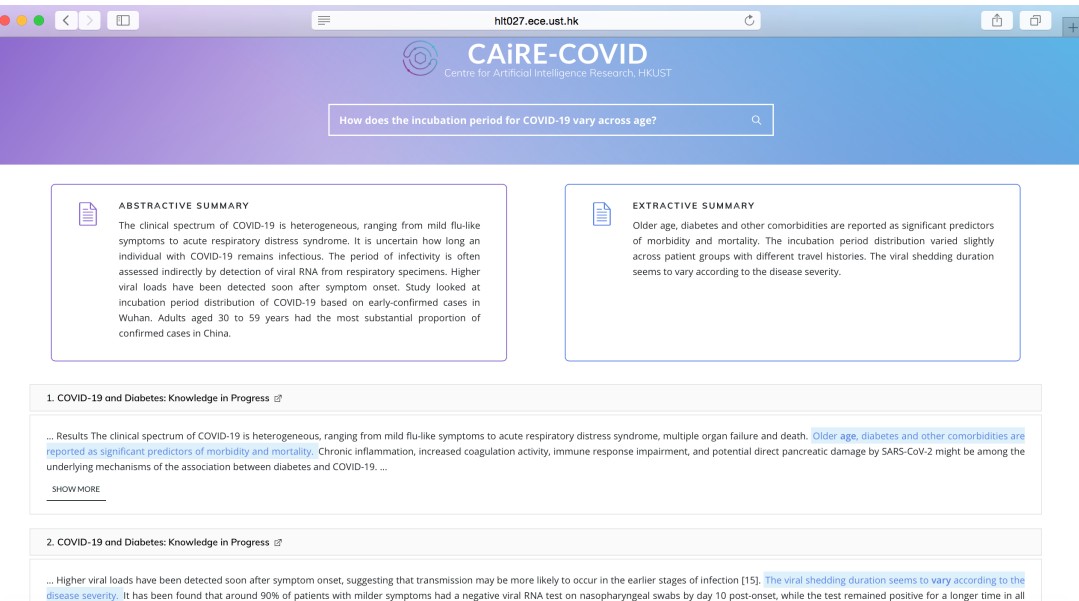

Figure 1: The user interface of our CAiRE-COVID website.

The effectiveness of our system has been proved by winning one of the tasks in Round 1 of the Kaggle CORD-19 Challenge,[1] in which hundreds of submissions were evaluated with the help of medical researchers. We further conduct a series of experiments to quantitatively show the competency of each module.

To enhance both generalization and domain-expertise capability, we use an ensemble of two QA models in the QA module as the evidence selector. We evaluate the performance on the recently released CovidQA (Tang et al., 2020) dataset, and the results indicate that our QA module even outperforms the T5 model (Raffel et al., 2019) on the recall metric, while for keyword questions, it also marginally outperforms T5 on the precision fraction.

The performance of the summerizer module is evaluated on two existing query-focused summarization (QFS) datasets, the DUC datasets (Dang, 2005; Hoa, 2006) and Debatepidia dataset (Nema et al., 2017), since there is no QFS dataset for COVID-19. The DUC datasets are the most widely used for the QFS task, while Debatepedia is the first large-scale abstractive QFS dataset. Previous works on the QFS task incorporate query relevance, either via a query-document relevance score (Baumel et al., 2018) or query attention model (Nema et al., 2017), into a seq2seq model, or concatenate query to documents into a pre-trained transformer architecture (Laskar et al., 2020; Savery et al., 2020). However, none have taken answer relevance into consideration. By incorporating answer relevance from the QA module into the summarization process, our query-focused multi-document summarizer achieves consistent ROUGE score improvement over the BART (Lewis et al., 2019)-based baseline method on the abstractive task, and the LEAD baseline on the extractive task on both datasets. Thus we believe that our proposed summarizer module can also work well on query focused summarization related to COVID-19 questions.

Furthermore, we have launched our CAiRE-COVID website (as shown in Figure 1), which enables real-time interactions for COVID-19-related queries by medical experts. The code[3] for our system is also open-sourced to help future study.

## 2 Related Work

With the release of the COVID-19 Open Research Dataset (CORD-19)[1] by the Allen Institute for AI, multiple systems have been built to assist both researchers and the public to explore valuable information related to COVID-19. CORD-19 Search[4] is a search engine that utilizes the CORD-19 dataset processed using Amazon Comprehend Medical. Google released the COVID19 Research Explorer a semantic search interface on top of the CORD-19 dataset. Meanwhile, Covidex[5] applies multi-stage search architectures, which can extract different

---

[3] https://github.com/HLTCHKUST/CAiRE-COVID
[4] https://cord19.aws/
[5] https://covidex.ai/

features from data. An NLP medical relationship engine named the WellAI COVID-19 Research Tool[6] is able to create a structured list of medical concepts with ranked probabilities related to COVID-19, and the tmCOVID[7] is a bioconcept extraction and summarization tool for COVID-19 literature.

Our system, in addition to information retrieval, gives high quality relevant snippets and summarization results given the user query. The website[2] further display information about COVID-19 in a well structured and concise manner.

# 3 Methodology

Figure 2 illustrates the architecture of the CAiRE-COVID system, which consists of three major modules: 1) Document Retriever, 2) Relevant Snippet Selector, and 3) Query-focused Multi-Document Summarizer.

## 3.1 Document Retrieval

To *select* the most relevant document, i.e. article or paragraph, given a user query, we first apply the Document Retriever with the following two sub-modules.

### 3.1.1 Query Paraphrasing

As shorter sentences are generally more easily processed by NLP systems (Narayan et al., 2017), the objective of this sub-module is to break down a user query and rephrase complex question sentences into several shorter and simpler queries that convey the same meaning. Its effectiveness has been proved in our CORD-19 Kaggle tasks, in dealing with the questions that are too long and complicated, and we show examples in Appendix B. Currently, this module has been excluded from our online system, since the automatic solutions we tried (Min et al., 2019; Perez et al., 2020) did not give satisfactory performance improvement for our system. More automatic methods will be explored in the future.

### 3.1.2 Search Engine

We use Anserin (Yang et al., 2018a) to create the search engine for retrieving a preliminary candidate set of documents. Anserini is an information retrieval module wrapped around the open source search engine Lucene[8] which is widely used to build industry standard search engine applications. Anserini uses the Lucene indexing to create an easy-to-understand information retrieval module. Standard ranking algorithms (e.g, bag of words and BM25) have been implemented in the module. We use paragraph indexing for our purpose, where each paragraph of the full text of each article in the CORD-19 dataset is separately indexed, together with the title and abstract. For each query, the module can return $n$ top paragraphs matching the query.

## 3.2 Relevant Snippet Selector

The Relevant Snippet Selector outputs a list of the most relevant answer snippets from the retrieved documents while highlighting the relevant keywords. To effectively find the snippets of the paragraphs relevant to a query, we build a neural QA system as an evidence selector given the queries. QA aims at predicting answers or evidences (text spans) given relevant paragraphs and queries. The paragraphs are further re-ranked based on a well-designed score, and the answers are highlighted in the paragraphs.

### 3.2.1 QA as Evidence Selector

**Evidence Selection** To enhance both generalization and domain-expertise capability, we leverage an ensemble of two QA models: the HLTC-MRQA model (Su et al., 2019) and the BioBERT (Lee et al., 2020) model. The HLTC-MRQA model is an XLNet-based (Yang et al., 2019) QA model which is trained on six different QA datasets via multi-task learning. This helps reduce over-fitting to the training data and enable generalization to out-of-domain data and achieve promising results. More details are mentioned in Appendix A. To adopt the HLTC-MRQA model as evidence selector into our system, instead of fine-tuning the QA model on COVID-19-related datasets, we focus more on maintaining the generalization ability of our system and conducting zero-shot QA.

To obtain a better performance, we also combine the HLTC-MRQA model with a *domain-expert*: the BioBERT QA model, which is fine-tuned on the SQuAD dataset.

**Answer Fusion** To increase the readability of the answers, instead of only providing small spans of answers, we provide the sentences that contain the predicted answers as the outputs. When the two QA models find different evidence from the same

---

[6]https://wellai.health/
[7]http://tmcovid.com/
[8]https://lucene.apache.org/

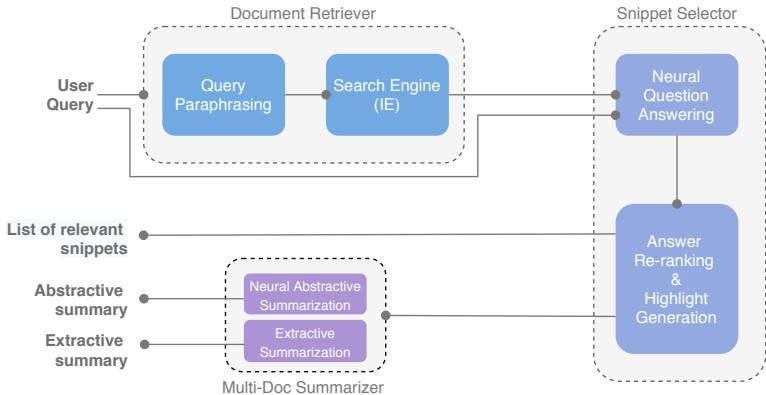

Figure 2: System architecture overview

paragraph, both pieces of evidence are kept. When the predictions from the two models are identical or there is an inclusion relationship between the two, the predictions will be merged together.

### 3.2.2 Answer Re-ranking and Highlight Generation

The retrieved paragraphs are further re-ranked based on the answer relevance to the query.

**Answer Confidence Score** We leverage the prediction probability from the QA models as the answer's confidence score. The confidence score of an ensemble of two QA models is computed as in Equation 1.

$$s_{conf} = \begin{cases} 0.5min\{|s_m|, |s_b|\} & if\, s_m, s_b < 0 \\ -max\{|s_m|, |s_b|\} & \\ s_m + s_b & otherwise, \end{cases}$$
(1)

where the confidence score from each model is annotated as $s_m$ and $s_b$.

**Keyword-based Score** We calculate the matching score between a query and the retrieved paragraphs based on word matching. To obtain this score, we first select important keywords from the query based on POS-tagging, only taking words with NN (noun), VB (verb), JJ (adjective) tags into consideration. By separately summing the term frequencies and the total number of important keywords that appear in the paragraph, we can get two matching scores, which are annotated as $s_{freq}$ and $s_{num}$, respectively. For the term-frequency matching score, we normalize shorter paragraphs using a sigmoid value computed from the paragraph length, and reward paragraphs with more diverse keywords from the query. The final matching score is com-

puted as in Equation 2.

$$s_{match} = \lambda_1 s_{freq} \cdot \sigma(l - l_c) + \lambda_2 s_{num},$$
(2)

where $l$ is the length of the paragraph and $l_c$ is a length constraint. Because of the effect of the sigmoid function, for data samples whose paragraph length is shorter or similar to $l_c$, the penalty will be applied to the final matching score.

**Re-rank and Highlight:** The re-ranking score is calculated based on both the matching score and the confidence score, as shown in Equation 3. The relevant snippets are then re-ranked together with the corresponding paragraphs and displayed via highlighting:

$$score_{re-rank} = s_{match} + \alpha s_{conf}.$$
(3)

### 3.3 Query-focused Multi-document Summarization

To efficiently present pertinent COVID-19 information to the user, we propose a query-focused multi-document summarizer to generate abstractive and extractive summaries related to COVID-19 questions.

#### 3.3.1 Abstractive Summarization

**BART Fine-tuning** Our abstractive summerization model is based on BART (Lewis et al., 2019), which obtained state-of-the-art results on the summarization tasks on the CNN/DailyMail datasets (Hermann et al., 2015) and XSUM (Narayan et al., 2018). We use the BART model fine-tuned on the CNN/DailyMail dataset as the base model since we do not have other COVID-19 related summarization data.

**Incorporating Answer Relevance** In order to generate query-focused summaries, we propose to

incorporate answer relevance in the BART-based summarization process in two aspects. First, instead of using the paragraphs list passed by the Document Retriever, we use the top $k$ paragraphs $\{para_1, para_2, .., para_k\}$ passed by the QA module as input to the Multi-document Summarizer, which are re-ranked according to their *answer relevance* to the query, as shown in Equation 3. Then, instead of using only the re-ranked answer-related paragraphs to generate a summary, we further incorporate *answer relevance* by concatenating the predicted answer spans from the QA models with each corresponding paragraph. We also concatenate the query to the end of the input, since this has been proved to be effective for the QFS task (Savery et al., 2020). So input to the summarization model is $C = \{\hat{para}_1, \hat{para}_2, .., \hat{para}_k\}$, where

$$\hat{para}_i = [para_i; ans\_spans_i; query] \quad (4)$$

**Multi-document Summarization**   Considering that each $\hat{para}_i$ in $C$ may come from different articles and focus on different aspects of the query, we generate the multi-document summary by directly concatenating the summary of each $\hat{para}$, to form our final answer summary. Some redundancy might be included, but we think this is fine at the current stage.

### 3.3.2   Extractive Summarization

In order to generate a query-focused extractive summary, we first extract answer sentences which contain the answer spans generated from the QA module, from multiple paragraphs as candidates. Then we re-rank and choose the top-$k$ (k=3) according to their answer relevance score to form our final summary. The *answer relevance* score is calculated in the following way:

**Sentence-level Representation**   To generate a sentence-level representation we sum the contextualized embeddings encoded by ALBERT(Lan et al., 2019), then divide by the sentence length. This representation can capture the semantic meaning of the sentence to a certain degree through a stack of self-attention layers and feed-forward networks. For a sentence with $n$ tokens $X = [w_1, w_2, .., w_n]$, the representation $h$ is calculated by Equation 5.

$$e_{1:n} = ALBERT([w_1, w_2, .., w_n])$$
$$h = \frac{\sum_{i=1}^{n} e_i}{n} \quad (5)$$

**Similarity Calculation**   After sentence representation extraction, we have embeddings for the an-

swer sentences and the query. In this work, the cosine similarity function is used for calculating the similarity score between them. For each query, only the top 3 answer sentences are kept.

## 4   Experiments

In Table 1 we show examples of each module. In order to quantitatively evaluate the performance of each module and show the effectiveness of our system, we conduct a series of respective experiments.

### 4.1   Question Answering

For the QA module, we conduct all the experiments with hyper-parameter $\lambda_1$ as 0.2, $\lambda_2$ as 10, $l_c$ as 50 and $\alpha$ as 0.5.

#### 4.1.1   Quantity Evaluation

**Dataset**   We evaluate our QA module performance on the CovidQA dataset, which was recently released by Tang et al. (2020) to bootstrap research related to COVID-19. The CovidQA dataset consists of 124 question-article pairs related to COVID-19 for zero-shot evaluation on transfer ability of the QA model.

**Experiment Settings**   The evaluation process on the CovidQA dataset is designed as a text ranking and QA task. For one article which contains $M$ sentences, we split it into $N(N < M)$ paragraphs. One sentence is selected as the evidence to the query from each of the paragraphs. The re-ranking scores for each sentences are meanwhile calculated. After evidence selection, we re-rank the $N$ sentences according to the re-ranking score (§3.2.2). The QA results are evaluated with Mean Reciprocal Rank (MRR), precision at rank one (P@1) and recall at rank three (R@3). However, in our case, MRR is computed by:

$$MRR = \frac{1}{|Q|} \sum_{i=1}^{|Q|} \{\frac{1}{rank_i}, 0\}, \quad (6)$$

where $rank_i$ is the rank position of the first sentence where the golden answer is located given one article (We assume it as the golden sentence). If there's no golden sentence selected in the $N$ candidates, we assign the score of the data sample as zero.

**Analysis**   The results are shown in Table 2. We test our models on both natural language questions and keyword questions. Changes in the efficiency of different models indicate their preferences for different kinds of questions. The HLTC-MRQA

**Query**: What are the risk factors for COVID-19? (*from Task-2*)

**Answer**: "Our analysis suggests that cardiovascular and kidney diseases, obesity, and hypertension are significant risk factors for COVID-19 complications, as previously reported." (Yanover et al., 2020) "Some prognostic factors beyond old age have been identified: for example, an increased body mass index is a major risk factor for requiring respiratory assistance. Indeed, obesity combines several risk factors, including impaired respiratory mechanics, the presence of other comorbidities and inappropriate inflammatory responses, partly due to ectopic fat deposits." (Scheen et al., 2020) "The Center for Disease Control and Prevention (CDC) suggests that neurological comorbidities, including epilepsy, may be a risk factor for COVID-19, despite the lack of evidence." (Kuroda, 2020)

**Abstractive Summary**: Reliably identifying patients at increased risk for COVID-19 complications could guide clinical decisions, public health policies, and preparedness efforts. The prevalence of diabetes patients hospitalized in intensive care units for COVID-19 is two- to threefold higher. An increased body mass index is a major risk factor for requiring respiratory assistance. The Center for Disease Control and Prevention (CDC) suggests that neurological comorbidities, including epilepsy, may be a risk factor for COVID-19. Presently, a medical history of epilepsy has not been reported.

**Extractive Summary**: The Center for Disease Control and Prevention (CDC) suggests that neurological comorbidities, including epilepsy, may be a risk factor for COVID-19, despite the lack of evidence. As such, it is unclear to what extent the prevalence of comorbidities in the studied population differs from that of same age (and sex) SARS-CoV-2 positive patients; and, accordingly, whether these comorbidities are significant risk factors for severe COVID-19 or merely a reflection of comorbidity prevalence in the wider population. What are the factors, genetic or otherwise, that influence interindividual variability in susceptibility to COVID-19, its severity, or clinical outcomes?

**Query**: What has been published about information sharing and inter-sectoral collaboration? (*from Task-10*)

**Answer**: "However, internal and external assessments and evaluations within both sectors indicate the persistence of specific gaps in the implementation of Joint Circular 16 on coordinated prevention and control of zoonotic diseases, information sharing and inter-sectoral collaboration." (Springer, 2016) "For example, our findings suggest that a key determining factor relating to cross-border collaboration is whether or not the neighbour in question is a fellow member of the EU. As a general rule, collaboration and information exchange is greatly facilitated if it takes place between two EU Member States as opposed to between an EU Member State and a non-EU Member State." (Kinsman et al., 2018) "Several system leaders called for further investment in knowledge sharing among a broad network of health system leaders to help advance the population health agenda:It would be great to have a consortium, a collaboration, some way to be able to do information sharing, maybe a clearing house . . . or even to formally meet to discuss and hear about and share successes . . . (CEO, Regional/District Health Authority)." (Cohen et al., 2014)

**Abstractive Summarization**: Epidemiology and laboratory collaboration between the human health and animal health sectors is a fundamental requirement and basis for an effective One Health response. During the past decade, there has been significant investment in laboratory equipment and training. For example, a key determining factor relating to cross-border collaboration is whether or not the neighbour in question is a fellow member of the EU. Several system leaders called for further investment in knowledge sharing among a broad network of health system leaders.

**Extractive Summarization**: Criteria selected in order of importance were: 1)severity of disease in humans, 2)proportion of human disease attributed to animal exposure, 3)burden of animal disease, 4)availability of interventions, and 5)existing inter-sectoral collaboration. Various rules-in-use by actors for micro-processes (e.g. coordination, information sharing, and negotiation) within NPGH arenas establish ranks and relationships of power between different policy sectors interacting on behalf of the state in global health. For example, our findings suggest that a key determining factor relating to cross-border collaboration is whether or not the neighbour in question is a fellow member of the EU.

Table 1: Example QA pairs and the abstractive and extractive summaries output given CORD-19[1] task questions from our system.

| Model | NL Question | | | Keyword Question | | |
|---|---|---|---|---|---|---|
| | P@1 | R@3 | MRR | P@1 | R@3 | MRR |
| T5(+ MS MARCO)[†] | 0.282 | 0.404 | 0.415 | 0.210 | 0.376 | 0.360 |
| BioBERT | 0.177 | 0.423 | 0.288 | 0.162 | 0.354 | 0.311 |
| HLTC-MRQA | 0.169 | 0.415 | 0.291 | 0.185 | 0.431 | 0.274 |
| Ensemble | 0.192 | **0.477** | 0.318 | **0.215** | **0.446** | 0.329 |

Table 2: Results of the QA models on the CovidQA dataset. [†]The T5 model (Raffel et al., 2019) which is fine-tuned on the MS MARCO dataset (Nguyen et al., 2016) is the strongest baseline from Tang et al. (2020). However, due to the difference in experiment settings, the MRR values from our models and those from baseline models are not comparable.

model with keyword questions shows better performance on precision and recall fractions, while the model with natural language questions is more likely to have relevant answers with a higher rank. The BioBERT model, however, performs under a different scheme. After making an ensemble of two QA models, the performance in terms of precision, recall and MRR fractions is improved. Moreover, our QA module even outperforms the T5 (Raffel et al., 2019) baseline on the recall metric, while

| Model Setting | ROUGE-1 | | | ROUGE-2 | | | ROUGE-L | | |
|---|---|---|---|---|---|---|---|---|---|
| | Recall | Precision | F1 | Recall | Precision | F1 | Recall | Precision | F1 |
| BART(C) | 19.60 | 8.80 | 11.80 | 3.22 | 1.41 | 1.91 | 16.76 | 8.17 | 10.70 |
| BART(C,Q) | 20.43 | 9.27 | 12.36 | 3.56 | 1.60 | 2.13 | 17.50 | 8.58 | 11.19 |
| BART(Q,C) | 19.16 | 8.49 | 11.43 | 3.06 | 1.31 | 1.76 | 16.39 | 7.77 | 10.25 |
| BART(A,Q) | 20.15 | 8.93 | 12.04 | 3.37 | 1.43 | 1.95 | 17.29 | 8.25 | 10.88 |
| BART(Q,A) | 19.15 | 8.57 | 11.48 | 2.97 | 1.27 | 1.70 | 16.46 | 7.88 | 10.36 |
| **BART(C,A,Q)** | **21.92** | **10.05** | **13.32** | **4.21** | **1.85** | **2.47** | **19.09** | **9.36** | **12.18** |

Table 3: Results for Debatepedia QFS dataset

| Model Setting | DUC 2005 | | | DUC 2006 | | | DUC 2007 | | |
|---|---|---|---|---|---|---|---|---|---|
| | 1 | 2 | SU4 | 1 | 2 | SU4 | 1 | 2 | SU4 |
| LEAD | 33.35 | 5.66 | 10.88 | 32.10 | 5.30 | 10.40 | 33.40 | 6.50 | 11.30 |
| Our Extractive Method | **35.19** | **6.28** | **11.61** | **34.46** | **6.51** | **11.23** | **35.31** | **7.79** | **12.07** |
| BART(C_$nr$) | 32.41 | 4.62 | 9.86 | 35.78 | 6.25 | 11.37 | 37.87 | 8.11 | 12.96 |
| BART(C) | 34.25 | 5.60 | 10.88 | 37.99 | 7.64 | 12.81 | 40.66 | 9.33 | 14.43 |
| BART(C,Q) | 34.20 | **5.77** | 10.88 | 38.26 | 7.75 | **12.95** | **40.74** | **9.60** | **14.63** |
| BART(C,A) | 34.29 | 5.70 | 10.93 | 38.31 | 7.60 | 12.90 | 40.71 | 9.11 | 14.30 |
| **BART(C,A,Q)** | **34.64** | 5.72 | 11.04 | **38.31** | **7.70** | 12.88 | 40.53 | 9.24 | 14.37 |

Table 4: Results for DUC datasets

for keyword questions, our model also marginally outperforms T5 on the precision fraction.

### 4.1.2 Case Study

Despite the fact that two models select the same sentence as the final answer given a question in most of the times when there is a reasonable answer in the paragraph, we observe that two models show different *taste* on language style. Figure 3 shows a representative example of QA module output. The prediction of the BioBERT model shows its preference for an experimental style of expression, while the prediction of the MRQA model is more neutral to language style.

### 4.2 Query-focused Multi-document Summarization

In order to generate query-focused summarizatioin for COVID-19 questions, we propose to incorporate answer relevance with the help of a QA model into the summarization process.

### 4.2.1 Datasets

Since there are no existing QFS datasets for COVID-19, we choose the following two datasets to evaluate the performance of the summarizer.

**DUC Datasets**  DUC 2005 (Dang, 2005) first introduced the QFS task. This dataset provides 50

queries paired with multiple related document collections. Each pair, has 4-9 human written summaries. The excepted output is a summary within 250 words for each document collection that can answer the query. DUC 2006 (Hoa, 2006) and DUC 2007 have a similar structure. We split the documents into paragraphs within 400 words to fit the QA model input requirement.

**Debatepedia Dataset**  This dataset is included in our experiments since it is very different from the DUC QFS datasets. Created by (Nema et al., 2017), it is the first large-scale QFS dataset, consisting of 10,859 training examples, 1,357 testing and 1,357 validation samples. The data come from Debatepedia, an encyclopedia of pro and con arguments and quotes on critical debate topics, and the summaries are debate key points that are a single short sentence. The average number of words in summary, documents and query is 11.16, 66.4, and 10 respectively.

### 4.2.2 Model Setting

**Abstractive Model Setting**  We use BART (Lewis et al., 2019) fine-tuned on XSUM (Narayan et al., 2018) as the abstractive base model for the Debatepedia dataset, since XSUM is the most abstractive dataset containing the highest number of novel bi-grams. Meanwhile, we use BART fine-tuned on CNN/DM for the DUC dataset to generate

Figure 3: An example of QA output of our system. The output of the QA module is highlighted in the paragraph in blue. We also use purple and blue underlining to distinguish the outputs of the HLTC-MRQA model and the BioBERT model.

longer summaries. Different input combination settings are tested.

*BART(C)*: We use the context only as the input to the BART model.

*BART(C,Q)*: We use the concatenation of the context and query as input to the BART model.

*BART(Q,C)*: We concatenate the query at the beginning of the context as the input to the BART model.

*BART(A,Q)*: We concatenate the answer sentences (sentences from the context that contain the answer spans) with the query as input to the BART model.

*BART(Q,A)*: We switch the position of query and answer sentences as input to the BART model.

*BART(C_nr)*: We use the context only as the input to the BART model. However, we do not re-rank the paragraphs in the context.

**BART(C,A,Q)**: We concatenate the context, answer spans, and query as input, which is the input configuration we adopt in our system.

For the DUC datasets, which contain multiple documents as context, we iteratively summarize the paragraphs which are re-ranked by the QA confidence scores till the budget of 250 words is achieved.

**Extractive Model Setting** We conduct extractive summarization on the DUC datasets. LEAD is our baseline (Xu and Lapata, 2020). For each document collection, LEAD returns all leading sentences of the most recent document up to 250 words. Our answer relavance driven extractive method has been introduced.

### 4.2.3 Results

We use ROUGE as the evaluation metric for the performance comparison. Table 3 and Table 4 show the results for the Debatepedia QFS dataset and DUC datasets respectively. As we can see from the two tables, by incorporating the answer relevance, consistent ROUGE score improvements of **BART(C,A,Q)** over all other settings are achieved on both datasets, which proves the effectiveness of our method. Furthermore, as shown in Table 4, consistent ROUGE score improvements are obtained by our extractive method over the LEAD baseline, and in the abstractive senario, BART(C) also outperforms BART(C_nr) by a good margin, showing that re-ranking the paragraphs via their answer relevance can help improve multi-document QFS performance.

## 5 Conclusion

In this paper, we propose a general system, CAiRE-COVID, with open-domain QA and query focused multi-document summarization techniques for efficiently mining scientific literature given a query. The system has shown its efficiency on the Kaggle CORD-19 Challenge, which was evaluated by medical researchers, and a series of experimental results also proved the effectiveness of our proposed methods and the competency of each module. The system is also easy to be generalized to general domain-agnostic literature information mining, especially for possible future pandemics. We have launched our website[2] for real-time interactions and released our code[3] for broader use.

## Acknowledgments

We would like to thank Yongsheng Yang, Nayeon Lee and Chloe Kim for their help in launching our CAiRE-COVID website.

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

# A Question Answering Module

## A.1 Details of HLTC-MRQA Model

The MRQA model (Su et al., 2019) is leveraged in the CAiRE-Covid system. To equip the model with better generalization ability to unseen data, the MRQA model is trained in a multi-task learning scheme on six datasets: SQuAD (Rajpurkar et al., 2016), NewsQA (Trischler et al., 2017), TriviaQA (Joshi et al., 2017), SearchQA (Dunn et al., 2017), HotpotQA (Yang et al., 2018b) and NaturalQuestions (Kwiatkowski et al., 2019). The training sets vary from each other in terms of data source, context lengths, whether multi-hop reasoning is needed and strategies for data augmentation. To evaluate the generalization ability, the authors utilized the BERT-large model (Devlin et al., 2019), which is trained with the same method as the MRQA model as the baseline. The models are evaluated on twelve unseen datasets, including DROP (Dua et al., 2019) and TextbookQA (Kemb-havi et al., 2017). From Table 5, the MRQA model consistently outperforms the baseline and achieves promising results on the QA samples, which are different from the training samples in terms of data resource, domain etc., including biomedical unseen datasets, such as BioASQ (Tsatsaronis et al., 2012) and BioProcess (Berant et al., 2014).

| Datasets | MRQA model | | Baseline | |
|---|---|---|---|---|
| | EM | F1 | EM | F1 |
| DROP | 41.04 | 51.11 | 33.91 | 43.50 |
| RACE | 37.22 | 50.46 | 28.96 | 41.42 |
| DuoRC | 51.70 | 63.14 | 43.38 | 55.14 |
| BioASQ | 59.62 | 74.02 | 49.74 | 66.57 |
| TQA | 55.50 | 65.18 | 45.62 | 53.22 |
| RE | 76.47 | 86.23 | 72.53 | 84.68 |
| BioProcess | 56.16 | 72.91 | 46.12 | 63.63 |
| CWQ | 54.73 | 61.39 | 51.80 | 59.05 |
| MCTest | 64.56 | 78.72 | 59.49 | 72.20 |
| QAMR | 56.36 | 72.47 | 48.23 | 67.39 |
| QAST | 75.91 | 88.80 | 62.27 | 80.79 |
| TREC | 49.85 | 63.36 | 36.34 | 53.55 |

Table 5: Results of the MRQA model on unseen datasets (Su et al., 2019). *TQA*, *RE* and *CWQ* are, respectively, the abbreviations for *TextbookQA*, *RelationExtraction* and *ComplexWebQuestions*.

# B Query Paraphrasing

In our Kaggle task, the queries are always long and complex sentences. In this case, splitting and simplification is needed. Here, we show examples of the original task queries and their corresponding para-phrased sub-questions:

**Task Question 1:** What the literature reports about Range of incubation periods for the disease in humans (and how this varies across age and health status)?

- What does the literature report about range of incubation periods for COVID-19 in humans?
- How does the range of incubation periods for COVID-19 vary across human health status?
- How does the range of incubation periods for COVID-19 vary across human age?

**Task Question 2:** What the literature reports about the evidence that livestock could be infected and serve as a reservoir after the epidemic appears to be over?

- What does the literature report about the evidence that livestock could be infected by COVID-19?
- How does the infected livestock serve as a COVID-19 reservoir after the epidemic appears to be over?

**Task Question 3:** What the literature reports about access to geographic and temporal diverse sample sets to understand geographic distribution and genomic differences, and determine whether there is more than one strain in circulation?

- What does the literature report about access to geographic sample sets of COVID-19?
- What does the literature report about access to temporal sample sets of COVID-19?
- What does the literature report about geographic-time distribution of COVID-19?
- What does the literature report about number of strains of COVID-19 in circulation?