# OpenReview forum: "CAiRE-COVID: A Question Answering and Query-focused Multi-Document Summarization System for COVID-19 Scholarly Information Management"
_EMNLP/2020/Workshop/NLP-COVID — NLP-COVID19-EMNLP Oral_

### Official Review · AnonReviewer3 · 2020-09-17
**Impressive and promising system**

**Rating:** 9
**Confidence:** 5

**Review:**

This paper describes the CAiRE-COVID QA and query-focused multi-document summarization system for COVID-19 literature. The system links together several SOTA models in QA and summarization in a novel and effective way. Overall, the modeling decisions are well justified, and the performance of the system is quite impressive.

A few comments / questions:

* The authors do not remark on factuality in abstractive summarization, though it would seem to be an important consideration here due to the domain being of healthcare and public health consequence.
* Related: for both extractive and abstractive multi-document summarization, especially in the scientific domain, contradictions can be a problem. How are the authors considering addressing this? Keeping track of sources as the authors have done for extractive summarization gives users a way of keeping track of this information, but what about in the abstractive case?

---

### Official Review · AnonReviewer2 · 2020-09-25
**Great work and presentation**

**Rating:** 9
**Confidence:** 4

**Review:**

Authors present question answering and query focused multi-document summarization techniques for mining scientific literature given a query.

The paper is very well-written with enough detail across the pipeline comprising three modules: 1) Document Retriever, 2) Relevant Snippet Selector, and 3) Query-focused Multi-Document Summarizer. They provide good evaluation with baselines and analysis. Open-sourcing the code along with clear instructions is commendable. Related work section could shed additional light on similar approaches or settings, but is understandable given the page lmits. Have authors interrogated cascading errors due to the nature of the pipeline? It’d be good to briefly comment on that

---

### Official Review · AnonReviewer1 · 2020-09-25
**Looks promising but there are concerns regarding evaluation**

**Rating:** 6
**Confidence:** 5

**Review:**

This paper presents a system designed to help medical professionals to quickly get short answers about COVID-19 based on multiple source documents. The system took part in a related Kaggle competition and won one of the tracks, which means that the answers for one particular question regarding Covid (What has been published about information sharing and inter-sectoral collaboration?) were considered to be the most relevant by medical experts. The system consists of a three modules: Document Retriever, which includes Query Paraphrasing and Search Engine (Anserini), Relevant Snippet Selector (HLTC-MRQA and BioBERT) and Multi-Document Summarizer. The authors released their tool and source code, which is undoubtedly very helpful for both medical professionals and NLP community working on similar tasks. The search/snippet extraction/ranking part follows an established pipeline, and the authors improved the extraction results by using the ensemble of a domain-specific model (BioBERT) with generic one (HLTC-MRQA). However, the parts of the system which sound more novel raise some concerns.
1. The Query Paraphrasing subsystem - the authors paraphrased the Kaggle task questions manually to improve the retrieval results, since, as they say, automatic question simplification methods did not help to improve the retrieval. One concern is that such paraphrasing could have introduced bias for Kaggle questions and is obviously not scalable for arbitrary questions; another question is why would you call it a “subsystem” instead of stating that the original questions were too complicated for the system to retrieve meaningful results?
2. The summarisers (both extractive and abstractive) are evaluated on datasets not related to COVID-19 or medical literature datasets. Though it is true that there is no multi-document QFS dataset for COVID-19, the authors could have used a medical QFS dataset such as Mollá, Diego, and Maria Elena Santiago-Martinez. “Development of a corpus for evidence based medicine summarisation.” (2011). [https://sourceforge.net/projects/ebmsumcorpus/], which would give a better indication of summarisation performance than unrelated news/debates datasets. Moreover, though the suggested extractive and abstractive models improve over the baselines of Lead and vanilla BART, the reported results for DUC 2005, 2006, 2007 datasets are worse than that of old non-neural extractive models [see, for example, (Ye, S., Chua, T. S., Kan, M. Y., & Qiu, L. (2007). Document concept lattice for text understanding and summarization. Information Processing & Management, 43(6), 1643-1662.] and recent neural abstractive models [for instance, (Zhu, Haichao, Li Dong, Furu Wei, Bing Qin, and Ting Liu. “Transforming Wikipedia into Augmented Data for Query-Focused Summarization.” arXiv preprint arXiv:1911.03324 (2019)].
3. There is no qualitative evaluation of the results apart from the claim that “effectiveness of the system has been proved by winning one of the tasks”. Unfortunately, it’s just one task out of ten where the authors knew the questions and pre-processed them to achieve better results. If one checks the results of the system for other questions, say “What do we know about asymptomatic transmission”, 4 out of 6 sentences of the abstractive summary have no relation to the question but rather talk about paediatric patients and PTSD. If quantitive evaluation against the COVID corpus was impossible, the authors could have performed a qualitative evaluation of relevance/factuality of the resulting summaries in line with recent tendencies to evaluate factuality instead of relying on ROUGE scores.
4. A suggestion rather than criticism - the results could have probably be improved by incorporating more biomed-related resources, for example, by fine-tuning on PubMed or CovidSum dataset instead of CNN/DailyMail